# OpenReview forum: "Investigating VLM Hallucination from a Cognitive Psychology Perspective: A First Step Toward Interpretation with Intriguing Observations"
_ICLR.cc/2026/Conference — Submitted to ICLR 2026_

### Official Review · Reviewer_SAF5 · 2025-10-25

**Soundness:** 3
**Presentation:** 3
**Contribution:** 2
**Rating:** 6
**Confidence:** 4

**Summary:**

This paper investigates VLM hallucinations from a cognitive psychology perspective, introducing a taxonomy of biases including sycophancy and "authority bias." The authors present AIpsych, a scalable benchmark with 60,000 image-question pairs designed with "trap options" to probe these behaviors. Through a multi-step prompting methodology, the paper classifies model failures into different bias types. Experiments on SOTA VLMs suggest that as model size increases, sycophancy (specifically Type II) increases while authority bias decreases. The results are also compared against a human subject study.

**Strengths:**

1.  **Good Perspective:** The paper's reframing of VLM hallucination using concepts from cognitive psychology (authority bias, sycophancy) is a valuable contribution, moving the discussion beyond simple error quantification.
2.  **Systematic Benchmark Design:** The AIpsych benchmark is well-designed. Its multi-step query process to differentiate *why* a model fails (e.g., knowing compliance vs. blind trust) is a clever methodology for interpreting model behaviors.
3.  **Interesting Scaling Findings:** The core finding that authority bias and sycophancy scale differently with model size (one decreasing, the other increasing) is an intriguing and important observation for the field.
4.  **Human-AI Comparison:** The inclusion of a human subject study provides a useful baseline and enhances the psychological framing of the paper.

**Weaknesses:**

1.  The concept of "authority bias" (over-trusting the user)  strongly overlaps with the "counterfactual presupposition" phenomenon discussed in prior work [1]. The paper fails to discuss or differentiate its contribution from this existing work.
2.  The benchmark's reliance on COCO 2014 validation and Visual Genome datasets introduces a unaddressed risk of data contamination, as these sets are often part of VLM training data. Furthermore, the manual quality check of only 200 images seems insufficient to guarantee the quality and lack of ambiguity for 60,000 questions.
3.  The evaluation of authority bias is purely categorical. The analysis could be much stronger by incorporating metrics of model uncertainty (e.g., confidence-weighted accuracy as in [1]), which would differentiate between low-confidence compliance and high-confidence belief in the flawed prompt.
4.  The study does not analyze the prompt sensitivity of the key instruction "Don’t be sycophantic"  used in the third prompt. The paper's sensitivity test only covers stylistic/lexical changes, not structural changes (e.g., placement, system vs. user prompt). It also fails to consider cases where this instruction might negatively cause a model to switch from a correct to an incorrect answer.
5.  The paper's finding that VLMs become more sycophantic with size appears to contradict findings from [1], which observed that larger models have an improved ability to detect prompt traps. This significant contradiction with related work is not discussed.

REF:

[1] Antidote: A Unified Framework for Mitigating LVLM Hallucinations in Counterfactual Presupposition and Object Perception

[2] CertainlyUncertain: A Benchmark and Metric for Multimodal Epistemic and Aleatoric Awareness

**Questions:**

Please see the weakness section above.

---

> ### Author Response · Authors · 2025-11-26
> **Discussion with Reviewer SAF5 - Part 1**
>
> We sincerely thank the reviewer for their time and effort in reviewing the paper. Please see our response below.
>
> **W1: The concept of "authority bias" (over-trusting the user) strongly overlaps with the "counterfactual presupposition" phenomenon discussed in prior work [1]. The paper fails to discuss or differentiate its contribution from this existing work.**
>
> We thank the reviewer for bringing the Antidote [1] paper to our attention. As this work was released on arXiv, which was concurrent with our timeline, we had not included it in our initial review. We have added a **discussion of Antidote to our Related Work section**.
>
> Below, we clarify the critical distinctions between our frameworks:
>
> First, to differentiate the core terminology we used. Counterfactual Presupposition Question (CPQ) in Antidote focuses on Object Hallucination. **A CPQ implicitly assumes the existence of a non-existent object**. The challenge here is the detection of non-existence. Whereas our study focuses on attribute grounding. **We test the model’s response to existing objects where the user provides a manipulated/false description**. This is a more challenging test of cognitive conflict: the model sees the object but must choose between its visual perception and the "authority" of the user's false input.
>
> Second, like many prior works, **Antidote primarily focuses on the frequency (accuracy)** of hallucination to validate a mitigation method. Our Approach focuses on cognitive diagnostics. **Rather than just counting errors, we map what is happening. We analyze behavioural trends across model families, offering a psychological taxonomy that explains the "root cause" of the error rather than just the outcome.**
>
> Third, Antidote evaluates 16 models. We provide a significantly broader evaluation of 32 models across varied architectures and sizes. This allows us to derive scaling laws and architectural insights that smaller sample sizes miss.
>
> Finally, our work serves as a diagnostic step toward AGI. We isolate the specific psychological tendency (Authority Bias) where models yield to authority—a critical "human-like" flaw that requires distinct measurements from standard hallucination benchmarks.
>
> **W2: The benchmark's reliance on COCO 2014 validation and Visual Genome datasets introduces a unaddressed risk of data contamination, as these sets are often part of VLM training data. Furthermore, the manual quality check of only 200 images seems insufficient to guarantee the quality and lack of ambiguity for 60,000 questions.**
>
> We acknowledge that popular datasets are almost likely included in the pre-training of the VLMs. However, we respectfully argue that this fact strengthens the significance of our findings rather than undermining them.
>
> If "data contamination" were the primary driver of model behaviour, **we would expect the models to remember the ground truth of these images and achieve near-perfect accuracy. Instead, our results show that models frequently behave like Authority Bias and Sycophancy, failing to answer correctly even on images they have likely seen during training.** This demonstrates that the cognitive bias induced by the prompt is strong enough to override both the visual encoder's immediate perception and the model's parametric memory of the training data. This confirms that these biases represent fundamental behavioural failures in reasoning, not merely a lack of domain knowledge.
>
> We apologize for the ambiguity in our initial description of the dataset sampling. To clarify: we **selected 200 images from each of the two source datasets (a total of 400 images)**. This sample represents approximately 13% of the total available data pool used for this specific evaluation. We have revised the new version to present these statistics clearly and prevent future confusion.
>
> **W3: The evaluation of authority bias is purely categorical. The analysis could be much stronger by incorporating metrics of model uncertainty (e.g., confidence-weighted accuracy as in [1]), which would differentiate between low-confidence compliance and high-confidence belief in the flawed prompt.**
>
> We would like to clarify the use of “confidence-weighted accuracy” in Antidote [1] with the reviewer. We did not find the content relevant to “confidence” and “confidence-weighted accuracy” in that paper. Antidote uses standard F1 metrics that involve Accuracy, Precision, and Recall.

---

> ### Author Response · Authors · 2025-11-26
> **Discussion with Reviewer SAF5 - Part 2**
>
> **W4: The study does not analyze the prompt sensitivity of the key instruction "Don’t be sycophantic" used in the third prompt. The paper's sensitivity test only covers stylistic/lexical changes, not structural changes (e.g., placement, system vs. user prompt). It also fails to consider cases where this instruction might negatively cause a model to switch from a correct to an incorrect answer.**
>
> We are conducting more prompt analysis, such as phrase replacement, and will update them once they are ready.
>
> We would respectively disagree with “It also fails to consider cases where this instruction might negatively cause a model to switch from a correct to an incorrect answer.” The prompt sensitivity test aims to test how sensitive the models are to changes in the prompts and justify the design of our benchmark. It is ok for the models to switch to an incorrect answer, and our observation suggests that this mostly happens in the small models.
>
> **W5: The paper's finding that VLMs become more sycophantic with size appears to contradict findings from [1], which observed that larger models have an improved ability to detect prompt traps. This significant contradiction with related work is not discussed.**
>
> We agree with the reviewer’s statement to a certain extent. Our results do not contradict Antidote’s results completely. Modern architectures such as Ovis, Qwen, and GPT perform better in trap identification. Whereas, outdated models such as LLaVA and InternVL contradict the findings.
>
> We have added a section (Line 407-411) that discusses the observation in our updated version.

---

> > ### Comment · Reviewer_SAF5 · 2025-11-28
> >
> > Thanks the authors for the detailed rebuttal. Having considered the authors' response as well as the comments from the other reviewers, I will keep my original score. I am in partial agreement with the other reviewers' perspectives and encourage the authors to incorporate their suggestions to enhance the paper.

---

### Official Review · Reviewer_vMxu · 2025-10-28

**Soundness:** 2
**Presentation:** 3
**Contribution:** 2
**Rating:** 2
**Confidence:** 4

**Summary:**

This paper studies hallucination in Vision-Language Models (VLMs) through a cognitive psychology lens. It proposes that model errors may mirror human-like biases such as authority bias and sycophancy. The authors build a benchmark, AIpsych, with 3,000 images and 60,000 automatically generated question–answer pairs to test these behaviors. Results on models like GPT-4o, Qwen2.5-VL, and LLaVA show that larger models tend to be more sycophantic but less authority-biased. A small human study provides limited comparison between model and human responses.

**Strengths:**

The paper raises an interesting interdisciplinary perspective, attempting to connect cognitive psychology with VLM hallucination analysis.
The writing is clear and the paper is well-structured, making it easy to follow the motivation and experimental design.

**Weaknesses:**

1; The mapping between model responses and human cognitive biases is analogical rather than theoretically defined. “Authority bias” closely overlaps with standard instruction following or over-alignment.
2: The study reports frequency trends without statistical testing or causal modeling. Observed scaling effects replicate well-known alignment patterns rather than revealing new mechanisms.
3. The benchmark mainly re-labels existing hallucination behaviors under psychological terms, no mitigation or modeling implications are provided.

**Questions:**

1. The paper reports that around 7–9% of GPT-generated object annotations or descriptions are inaccurate, and states that such errors “serve as natural traps rather than noise.” However, it is unclear whether these samples were explicitly retained in the benchmark, and whether their impact on evaluation consistency was empirically analyzed. For a benchmark study, it would be important to validate this assumption (e.g., by comparing model behaviors on verified vs. noisy samples) to justify that these inaccuracies indeed contribute meaningfully rather than introducing uncontrolled variance.
2. How are the four categories formally distinguished, and how robust are they to prompt phrasing or translation?
3. The discussion on architecture effects (Section 4) seems largely descriptive, e.g., claiming that Qwen2.5-based models “consistently outperform” others due to stronger language backbones. However, the paper does not analyze why these architectural differences lead to distinct cognitive patterns, nor does it control for confounding factors such as dataset size, instruction tuning, or parameter count. Could the authors provide a more systematic validation (e.g., matched-size ablation or cross-backbone comparison) to support the claim that backbone choice, rather than training scale or data, drives the observed psychological behaviors?

---

> ### Author Response · Authors · 2025-11-26
> **Discussion with Reviewer vMxu - Part 1**
>
> We sincerely thank the reviewer for their time and effort in reviewing the paper. Please see our response below.
>
> **W1: The mapping between model responses and human cognitive biases is analogical rather than theoretically defined. “Authority bias” closely overlaps with standard instruction following or over-alignment.**
>
> Our use of psychological taxonomy is deliberately analogical. We fully acknowledge that current models are next-token predictors rather than AGI agents with human-like reasoning capabilities (aligning with views such as Yann LeCun’s on the nature of LLMs). Consequently, **we use terms like 'Authority Bias' and 'Sycophancy' to provide a structured framework for categorizing cognitive biases that mirror human cognitive failures, even if they stem from different underlying mechanisms. Therefore, mechanical problems such as standard instruction following and over-alignment may be the cause of the cognitive bias rather than an alternative definition of “authority bias”.**  Our primary objective is to point a new and neglected direction to access the model’s human-like behaviours for optimizing the models currently deployed in society and the development of true AGI. This is crucial for preventing misguided outcomes in high-stakes applications where Authority Bias or Sycophancy behaviours could threaten lives or societal safety.
>
> The reviewer suggests "Authority Bias" overlaps with standard instruction following. However, our results demonstrate that these two phenomena trend in opposite directions, given that they have distinct mechanisms.  We observe that as model size increases, Authority Bias decreases, whereas Sycophancy increases. If Authority Bias were simply "standard instruction following/over-alignment," it should correlate positively with Sycophancy and model capability. The fact that it behaves inversely highlights that it captures a specific failure mode that essentially competes with the model's visual grounding, rather than simply reflecting its alignment training. Furthermore, **it would be more appropriate to describe cognitive bias (e.g. Authority Bias) as an outcome and “standard instruction following or over-alignment” is the cause. They are not independent or in conflict.**
>
> Notably, we offer the 4th prompt that provides the model with the correct option “Else”. The majority of the SOTA VLMs score less than a 20% chance to choose the correct choice, “Else”. Only Qwen2.5-VL make the exception, which highlights its unique strategies in countering the cognitive bias. Nevertheless, Qwen2.5-VL shows improvements, but its best model fails more than half of the time, suggesting there is much room for improvement.
>
> **W2: The study reports frequency trends without statistical testing or causal modeling. Observed scaling effects replicate well-known alignment patterns rather than revealing new mechanisms.**
>
> We respectfully disagree that the study lacks causal insight. While we do not employ structural causal models, **our multi-turn experimental design is explicitly interventional**. By actively manipulating the prompt context, we investigate the model’s generation process to isolate the cause of the error.
>
> The trends we report are observed consistently across 32 distinct models (Appendix A.3). The magnitude of the behavioural shifts provides strong empirical evidence that **these are systematic signal patterns, not random noise**. Also, as detailed in Appendix A.4, we conducted **extensive sensitivity analyses** on our reliability metrics to prove that our findings are **statistically robust**.
>
> **W3: The benchmark mainly re-labels existing hallucination behaviors under psychological terms, no mitigation or modeling implications are provided.**
>
> Our benchmark and ReS offer a new objective function for model selection that penalizes the specific behavioural drift that standard accuracy metrics miss. Our **primary goal is to provide a benchmark (AIpsych) to evaluate VLM behaviours as a step toward AGI**, identifying critical cognitive biases like Authority Bias. We acknowledge that providing a direct mitigation method is desirable. However, we emphasize that problems like sycophancy have long plagued the research community; despite numerous attempts and significant resource investment, these behaviours persist even in SOTA models.
>
> In this work, **we observe intriguing traits in SOTA VLMs that can help explain why previous attempts have struggled**. Specifically, our analysis of modern techniques (detailed in Appendix A.2) reveals that standard alignment methods often reduce Authority Bias only at the cost of exacerbating Sycophancy. Based on the results, we have some inspiration for future work.
>
> Due to the page limit, we have newly included the **Limitations and Future Work section in the updated paper to provide insights to solve or prevent the cognitive biases.**

---

> ### Author Response · Authors · 2025-11-26
> **Discussion with Reviewer vMxu - Part 2**
>
> **Q1: The paper reports that around 7–9% of GPT-generated object annotations or descriptions are inaccurate, and states that such errors “serve as natural traps rather than noise.” However, it is unclear whether these samples were explicitly retained in the benchmark, and whether their impact on evaluation consistency was empirically analyzed. For a benchmark study, it would be important to validate this assumption (e.g., by comparing model behaviors on verified vs. noisy samples) to justify that these inaccuracies indeed contribute meaningfully rather than introducing uncontrolled variance.**
>
> 	We fully agree with the reviewer’s comment that the validation of the annotation errors is important to justify our claim on “natural traps”.  We have conducted an ablation study on manually curated questions to **test GPT in Appendix A.1**.
>
> We have **newly conducted a set of studies in Appendix B.1 in the updated version**. The results show that the difference between the question with 9% of natural traps and no natural traps is **very small (change <1%)**. Also, the performance improvements switch between the two groups, suggesting the changes are possibly the result of intrinsic stochasticity in the model inference.
>
> Larger models require much higher computational power and longer inference time; we will update their results in the camera-ready version.
>
> **Q2: How are the four categories formally distinguished, and how robust are they to prompt phrasing or translation?**
>
> As described in **Section 3.1** in the first submission, we formally distinguish the four categories using a multi-turn question that is similar to the idea of a chain of thoughts. The first prompt is to verify if the model shows any cognitive bias. The second and third prompts help us to identify the type of cognitive bias.
>
> Also, **we have conducted the prompt sensitivity test in (Section 4 and Table 11) in the first submission**. In short, small models are susceptible to the prompt phrasing, but the large models are robust to it. Mover, we are conducting an additional test by replacing the position of instructions in the prompts.
>
> **Q3: The discussion on architecture effects (Section 4) seems largely descriptive, e.g., claiming that Qwen2.5-based models “consistently outperform” others due to stronger language backbones. However, the paper does not analyze why these architectural differences lead to distinct cognitive patterns, nor does it control for confounding factors such as dataset size, instruction tuning, or parameter count. Could the authors provide a more systematic validation (e.g., matched-size ablation or cross-backbone comparison) to support the claim that backbone choice, rather than training scale or data, drives the observed psychological behaviors?**
>
> We agree that a fully controlled ablation study (re-training models from scratch with controlled datasets and varying only the backbone) would be the ideal scientific standard to study the causal role of architecture.
> However, as the reviewer likely appreciates, reconstructing and fine-tuning SOTA VLMs to control for dataset size, instruction tuning recipes, and backbone architecture requires computational resources (hundreds of GPUs and proprietary datasets) that are generally **inaccessible to academic research**.
>
> While we cannot retrain the models, our **extensive analysis of 32 distinct models (detailed in Appendix A.3 in the first submission) **serves as a robust "natural experiment." We **compared models with matched/similar parameter counts** (e.g., comparing LLaVA-NeXT-7B, Qwen2.5-VL-7B, and InternVL3-8B). In these matched-size comparisons, the Qwen-based models consistently exhibited the best behaviours compared to counterparts of the same size.

---

### Official Review · Reviewer_gT1U · 2025-10-29

**Soundness:** 2
**Presentation:** 3
**Contribution:** 2
**Rating:** 2
**Confidence:** 4

**Summary:**

The paper aims to analyze hallucination phenomena in VLMs from a cognitive psychology perspective.
The authors introduce AIpsych, a benchmark designed to categorize VLM hallucinations into several types inspired by human cognitive biases (like authority bias, sycophancy, logical inconsistency). They further propose a ReS measue to quantify how strongly a model is influenced by each bias, conduct extensive experiments across multiple VLMs. The main claim is that VLM hallucinations share structural similarities with human cognitive biases, offering a psychologically grounded interpretation of hallucination behaviors.

**Strengths:**

- This paper is addressing one of important challenges towards the multi-modal AI systems. They re-organized the current hallucination problems in the VLMs' responses into cognitive psychology, which is novel angle for conceptual shift. They defined and curated psychologincal taxanomy for the VLM's bias and hierarchically designed for extend to AIpsych bnehcmark that systematically organizes diverse visual scenarios for the hallucination.

- The paper evluates multiple VLM of diffrent scales covering various architectures trained with various datasets, which also includes analysis with human study.

**Weaknesses:**

- The idea of linking cognitive bias with hallucination is interesting, but the data design does not really support this claim. Most tasks in the benchmark are simple binary or attribute-level questions that test visual recognition rather than cognitive reasoning. The results seem to reflect perceptual errors, not true psychological bias.

- As one of the critical issues is that the paper mainly describes and categorizes hallucination types without suggesting how to reduce or handle them. There is no discussion of mitigation methods or model improvements based on these findings.

**Questions:**

- How do the authors verify that their experiments actually measure cognitive bias rather than simple visual recognition errors or dataset artifacts?

- Can the proposed Reliability Score (ReS) be validated against human judgment or calibration metrics to show that it truly reflects bias instead of general uncertainty?

---

> ### Author Response · Authors · 2025-11-26
> **Discussion with Reviewer gT1U - Part 1**
>
> We sincerely thank the reviewer for their time and effort in reviewing the paper. Please see our response below.
>
> **W1： The idea of linking cognitive bias with hallucination is interesting, but the data design does not really support this claim. Most tasks in the benchmark are simple binary or attribute-level questions that test visual recognition rather than cognitive reasoning. The results seem to reflect perceptual errors, not true psychological bias.**
>
> **We respectfully disagree with the premise that these two factors are exactly comparable in this context.** It is inadequate to compare cognitive bias and visual recognition errors in this way, as we view **cognitive bias as a type of behaviour**, whereas **visual recognition error is a possible underlying cause** that can lead to such behaviour.
>
> While it is possible that imperfect visual representations captured by visual encoders are a factor contributing to bias, this does not negate the behavioural nature of the failure. We have updated our Limitations and Future Work section to explicitly discuss visual representation as a potential cause of the cognitive biases we measure.
>
> **W2: As one of the critical issues is that the paper mainly describes and categorizes hallucination types without suggesting how to reduce or handle them. There is no discussion of mitigation methods or model improvements based on these findings.**
>
> We frame our work as a necessary diagnostic prerequisite to effective mitigation. Our findings reveal that **current standards and best mitigation strategies cannot fix these hallucinations.** As detailed in Appendix A.2, existing de-biasing techniques, such as RLHF, focus on demographic or modality biases. Crucially, our experiments with QwenVL2.5 reveal that while scaling and alignment reduce Authority Bias, they exacerbate Sycophancy. This is a critical insight: it tells the researcher that simply scaling up or applying standard preference optimization trades one bias for another.
>
> Also, by differentiating 'Authority Bias' from 'Sycophancy', we can fix these problems more effectively. For example, future training can specifically enforce models not to agree when a user makes a false claim through prompt tuning.
>
> The newly added Limitations and Future Work section provides insights for mitigation methods.
>
> **Q1: How do the authors verify that their experiments actually measure cognitive bias rather than simple visual recognition errors or dataset artifacts?**
>
> The question is similar to Weakness 1. **The visual recognition error is a possible cause of the cognitive bias**. A future work to investigate their causality can delve into the attention map of the models, particularly the visual encoders.
>
> **Q2: Can the proposed Reliability Score (ReS) be validated against human judgment or calibration metrics to show that it truly reflects bias instead of general uncertainty?**
>
> We can validate the ReS by establishing a human baseline, treating human reasoning as the standard for non-hallucinatory or low-hallucinatory behaviour. As our survey result suggests, humans demonstrated a specific behavioural pattern: they rarely fell for Authority Bias (12.8%) or Type I Sycophancy (0.3%) compared to models, and frequently used the “Else” option (81%) to reject false manipulated premises.
>
> **The high frequency of “Else” option also implies that humans can accurately capture the visual information, and their responses were subconsciously influenced by the deliberately designed questions.** The ReS metric is designed to reward these human-like traits. It heavily penalizes Authority Bias and Sycophancy while rewarding the ability to spot traps. Consequently, as models improve and behave more human-like, their ReS increases, confirming that the metric correctly aligns with validated human judgment patterns. Hence, ReS is aligned with the goal of AGI, which is a machine with human-like cognitive abilities.

---

### Official Review · Reviewer_f7GR · 2025-11-01

**Soundness:** 2
**Presentation:** 1
**Contribution:** 2
**Rating:** 2
**Confidence:** 3

**Summary:**

The paper examines context bias (LLMs will believe things if they appear in context; also related to in-context learning) from the lens of sycophancy. To study these behaviors, the researchers developed a benchmark called AIpsych, which consists of 3,000 images and 60,000 questions designed to trap the models. They tested a variety of models as well as humans on this benchmark. They present the outcomes and some analysis of these outcomes.

**Strengths:**

- The paper examines the problems discussed from an interesting angle. I have to admit this is the first paper I've read that tries to fit human psychology concepts and anthropomorphize them this much. It remains to be seen whether this is the right approach but it is interesting at least.
- The benchmark introduced seems to be of reasonable size and I think it will provide insights on model behaviors.

**Weaknesses:**

- The writing is a bit poor (eg. at end of page 4: "because we want to see if the model naturally inherent some psychological disorder without intervention") and the presentation is generally confusing.

- The paper's central claim rests on a flimsy and unclear distinction between "sycophancy" and "authority bias." The authors essentially rename a known artifact of autoregressive models - bias towards information present in context - to "authority bias." From the context of a VLM responding to a flawed/broken user query, the difference between these two concepts is not obvious, and the methodology used to separate them is weak. It relies on the strong assumption that directly asking the model if there's a mistake and taking its "yes/no" response at face value is a faithful report of an internal "belief state," rather than the model simply succeeding or failing at a secondary meta-reasoning task.

- The point above is further confounded by the fact that human preference alignment can amplify sycophancy (user-pleasing behavior) but does not necessarily associate said behavior with the word/concept of sycophancy. I would the presented benchmark is measuring truthfulness or usefulness which are other metrics human preference alignment may optimize for, provided it is targeted in the model's post-training dataset.

- The paper does not seem to discuss the sampling settings used or log probabilities of model outputs. As an simple example: Assume the model outputs probabilities of 51% and 49% for "yes", "no" respectively. Naively sampling would either reduce the results to coin-tossing. Greedy sampling would ignore that the model was not very confident. More clarification on methodology here is needed.

- The paper does not seem to establish reliable baselines or references, and the proposed benchmark dataset seems to be entirely constructed of bad/corrupted questions. For example, it may be interesting to test the performance of the models on the task without any corruption of the prompt as this lets us have a reasonable reference/baseline to compare and normalize on. It is also relevant to test the "mistake verification" and "prevent sycophancy" portions of the prompt with clean questions to identify rate of hallucinations at those tasks.

- The authors seem to generate all of the dataset with GPT4o and then report GPT4o as an outlier but they do not seem to attempt to analyze or correct for this in any way (eg. generating another 60k questions would be cheap with Gemini Flash and testing GPT4o and other models on that may show whether or not GPT4o performs anomalously only on GPT4o generated prompts).

- Table 2 shows that 79% of human respondents did not provide a full response. Given the already small sample size, it is difficult to say whether there is anything conclusive about the human survey presented.

**Questions:**

- The paper makes multiple references to "GPT" with no further clarification on the specific model under question. Is it GPT-4V? GPT4o? GPT4o mini? Some GPT-5 variant? What is the model's date on the API?
- Figure 3 has "GPT" listed with 2 different points along the "parameter size" axis. As far as I know, there's not many reliable source for the actual parameter count or architecture (dense vs moe..) for OpenAI models. Clearly citing and clarifying what models are being discussed would be appreciated.

---

> ### Author Response · Authors · 2025-11-26
> **Discussion with Reviewer - f7GR Part1**
>
> We sincerely thank the reviewer for their constructive comments and effort. Please see our response below.
>
> **W1: The writing is a bit poor (eg. at end of page 4: "because we want to see if the model naturally inherent some psychological disorder without intervention") and the presentation is generally confusing.**
>
> Sorry for any confusion we caused; we have improved the phrasing in the updated version (Line 213-215).
>
> **W2: The paper's central claim rests on a flimsy and unclear distinction between "sycophancy" and "authority bias." The authors essentially rename a known artifact of autoregressive models - bias towards information present in context - to "authority bias." From the context of a VLM responding to a flawed/broken user query, the difference between these two concepts is not obvious, and the methodology used to separate them is weak. It relies on the strong assumption that directly asking the model if there's a mistake and taking its "yes/no" response at face value is a faithful report of an internal "belief state," rather than the model simply succeeding or failing at a secondary meta-reasoning task.**
>
> We refute the claim that the distinction is flimsy. If Authority Bias and Sycophancy were simply the same "context bias" artifact, they would scale identically. However, the experiments suggest that Sycophancy and Authority Bias are inversely correlated with model scale.
>
>   It may also be too rushed to conclude that our idea is to rename the “bias towards information present in context”. Every set of prompts has an “else” detour for the models, and almost every model scored less than 20% to choose the “else” option that includes GPT-4o. **The systematic rejection of the valid ``else'' option in favour of the false trap proves that the existing SOTA are not just attending to context, but specifically deferring to the information of the user's manipulated prompts. Notably, small models with a high Authority Bias rate are poor in instruction following**,  as shown in Table 10, further suggesting that the models are not just simply biased towards existing information. Importantly, **we would consider “bias towards information present in context” a cause, and Authority Bias is the consequence.**
>
> We agree that the methodology to separate the phenomena is not sophisticated. However, we would argue that most of the models, especially the small ones, do not even have the capability to answer the questions completely. This situation does not undermine our contribution; rather, our benchmark can tell that they are not reliable models directly. We believe that it is also our **future work to include more sophisticated prompts when the models become intelligent.**
>
> **W3: The point above is further confounded by the fact that human preference alignment can amplify sycophancy (user-pleasing behavior) but does not necessarily associate said behavior with the word/concept of sycophancy. I would the presented benchmark is measuring truthfulness or usefulness which are other metrics human preference alignment may optimize for, provided it is targeted in the model's post-training dataset.**
>
> Some of the models have been reinforced with humans’ preferences; the preference optimization may not explicitly involve truthfulness. **The ReS in the benchmark indeed can be modified as a loss to possibly improve a model’s reliability**. We will include this in future work.
>
>
> **W4: The paper does not seem to discuss the sampling settings used or log probabilities of model outputs. As an simple example: Assume the model outputs probabilities of 51% and 49% for "yes", "no" respectively. Naively sampling would either reduce the results to coin-tossing. Greedy sampling would ignore that the model was not very confident. More clarification on methodology here is needed.**
>
> We acknowledge the reviewer’s concern over the coin-tossing situation. The 4th Prompt provides an explicit "Else" option. If a model were simply uncertain between two trap options, a well-calibrated model could select "Else." The fact that models rarely use "Else" indicates they are not just "uncertain"—they are confident in the wrong answer. Meanwhile, we are conducting experiments on the models’ average confidence in the tokens.

---

> ### Author Response · Authors · 2025-11-26
> **Discussion with Reviewer - f7GR Part2**
>
> **W5: The paper does not seem to establish reliable baselines or references, and the proposed benchmark dataset seems to be entirely constructed of bad/corrupted questions. For example, it may be interesting to test the performance of the models on the task without any corruption of the prompt, as this lets us have a reasonable reference/baseline to compare and normalize on. It is also relevant to test the "mistake verification" and "prevent sycophancy" portions of the prompt with clean questions to identify rate of hallucinations at those tasks.**
>
> While we acknowledge the potential value of testing with uncorrupted prompts, we emphasize that our 4th Prompt condition explicitly serves as a valid 'escape hatch' by offering the 'Else' option. Critically, our results in Tables 7 and 9 demonstrate that despite having this valid path to reject the false premise, nearly all models selected the 'Else' option in less than 20% of cases. This systematic failure to utilize a truthful exit option—even when available—confirms that the observed bias is a behavioural preference for the trap, not merely a forced choice artifact.  We also newly added a section in Section 4 to explicitly discuss the results of the “else” option.
>
>
> **W6: The authors seem to generate all of the dataset with GPT4o and then report GPT4o as an outlier but they do not seem to attempt to analyze or correct for this in any way (eg. generating another 60k questions would be cheap with Gemini Flash and testing GPT4o and other models on that may show whether or not GPT4o performs anomalously only on GPT4o generated prompts).**
>
> We would like to clarify our description of GPT-4o to be an outlier. Our interpretation is that the GPT-4 models we tested may look like an outlier superficially and visually (given the scale of the plot).  As discussed in (Line 370-371), GPT-4o shows a little bit have Authority Bias rate (1.4%+), but it is offset by a much better trap spotting rate (+9%).
>
> Unfortunately, as part of an academic research group, we do not have the resources for Gemini. However, **we had an ablation study on GPT’s possible circularity problem using a total of 400 images with manually curated questions in Appendix A.1 in our first submitted version**. The results suggest that, being the generator of the benchmark, GPT4 models are not biased as their performance remains consistent in trend.
>
> **W7: Table 2 shows that 79% of human respondents did not provide a full response. Given the already small sample size, it is difficult to say whether there is anything conclusive about the human survey presented.**
>
> The 79% of respondents flagged as 'not providing a full response' were **fully valid** for analysis. These participants skipped the first part of the second and third prompts because they had already provided the answer in the first prompt. Some of the respondents make assumptions and intuitive decisions. Therefore, the data remains conclusive for measuring bias.
> While larger studies are always beneficial, our sample size is consistent with or exceeds standard benchmarks in this domain. For context, the **GPT-3 human evaluation [1] utilized only 80 participants**. Conducting a larger survey involves more resources, legal, and ethical issues. **We framed our work as  “a first step” toward bridging between human and machine cognition.** The goal is to encourage more researchers to work in the related interpretable and safe AI.
>
>
> **Q1: The paper makes multiple references to "GPT" with no further clarification on the specific model under question. Is it GPT-4V? GPT4o? GPT4o mini? Some GPT-5 variant? What is the model's date on the API?**
>
> Sorry for any confusion we caused. The benchmark is generated using GPT-4o (Line 101-120; 347-349). Evaluation of GPT models involves GPT-4o and GPT-4o-mini. We have clarified it in the updated version.
>
> **Q2:Figure 3 has "GPT" listed with 2 different points along the "parameter size" axis. As far as I know, there's not many reliable source for the actual parameter count or architecture (dense vs moe..) for OpenAI models. Clearly citing and clarifying what models are being discussed would be appreciated.**
>
> We apologize for the omission in the plot descriptions. We clarify that the models labelled "GPT" in the visualizations correspond to GPT-4o and GPT-4o-mini, consistent with the data presented in Tables 3, 6, 9, and 10. **We have updated the figure captions in the revised manuscript to explicitly state them.**
>
> There is no official public documentation confirming the exact parameter counts of the GPT-4o family. We have plotted them in a way that GPT-4o is the larger model compared to GPT-4o-mini. This ordering allows us to visualize performance trends relative to model capability, even if exact parameter scaling cannot be confirmed.
>
>
>
> [1] Brown, T., et al. “Language Models are Few-Shot Learners”, NeurIPS.

---

### Meta-Review · Area_Chair_mfpv · 2026-01-07

**Summary:**

Investigates whether VLM’s cognitive biases lead to hallucinations, which the authors categorize as  sycophancy, logical inconsistency, and “appeal to authority”. Introduces a benchmark, AIPsych to highlight these hallucination issues with current models. They also propose an ReS metric to quantify each bias for a model. The authors test 32 SOTA VLMs and conduct a human study, finding that larger models exhibit stronger “sycophantic tendencies” but “reduced authority bias”.

**Reviewer Concerns:**

**Addressed**
1. Some decisions have been made which demonstrate authors treating model specifications somewhat carelessly (e.g. GPT parameter count).
2. Related work released 5 months before the ICLR deadline was not originally taken into account, but was addressed in the rebuttal.
3. A claim about data contamination from reviewer saf5 was successfully addressed.

**Unaddressed**
1. Reviewers f7gr and vxmu point out that the distinction between sycophancy and authority bias is not clear. The authors answer this by simply stating that the empirical divergence between the results should settle the differences between the two metrics. This logic is not convincing since it justifies the hypothesis of the work simply via the authors’ own proposed empirical score!
2. The methodology for separating bias types is considered weak, relying on taking model yes/no responses "at face value" rather than measuring true internal states (saf5, f7gr). Critical details are missing: sampling settings, log probabilities, and baseline performance on uncorrupted prompts. The lack of statistical testing or causal modeling limits the strength of conclusions. It is also unclear why size differences cause the different biases, if the authors’ hypothesis is true. Authors keep addressing this by stating the importance of the “else” option, which misses the broader argument reviewers are making.
3. Reviewers (f7GR, vxmu, saf5) raise important questions about the quality of the benchmark, which could be addressed by examining the models’ response to the robustness of the prompt to paraphrasing or to explicit instructions, such as don’t be sycophantic. 7–9% of GPT-generated object annotations or descriptions are inaccurate, and states that such errors “serve as natural traps rather than noise.”
4. All questions are generated using GPT-4o and its own performance is not rigorously tested using generations from other models (does not need to be a paid model). The ablation study in Appendix A1 is hard to follow: how exactly were the image-question pairs manually generated?
5. Inconclusive human studies: Manual quality checking of only 200 images seems insufficient for 60,000 questions.
6. Anthropomorphizing model behavior using the lens of human psychology (f7GR, gt1u, vxmu) is an interesting idea, but the paper does not do enough to motivate model behavior on the basis of these cognitive biases we possess.

**Reviewer Scores:**

Only one of the reviewers responded back to the authors, keeping their original score. In my assessment, the authors response does not adequately address the concerns the other reviewers brought up.

---

### Decision · Program_Chairs · 2026-01-26

Reject